# Building a Real-Time Testing Platform for Unmanned Ground Vehicles with UDP Bridge

**DOI:** 10.3390/s22218493

**Published:** 2022-11-04

**Authors:** Łukasz Sobczak, Katarzyna Filus, Joanna Domańska, Adam Domański

**Affiliations:** 1Institute of Theoretical and Applied Informatics, Polish Academy of Sciences, Bałtycka 5, 44-100 Gliwice, Poland; 2Department of Distributed Systems and Informatic Devices, Faculty of Automatic Control, Electronics and Computer Science, Silesian University of Technology, Akademicka 16, 44-100 Gliwice, Poland

**Keywords:** SLAM, autonomous driving, AGV, simulation, UDP, Google Cartographer, ORB SLAM

## Abstract

Perception and vehicle control remain major challenges in the autonomous driving domain. To find a proper system configuration, thorough testing is needed. Recent advances in graphics and physics simulation allow researchers to build highly realistic simulations that can be used for testing in safety-critical domains and inaccessible environments. Despite the high complexity of urban environments, it is the non-urban areas that are more challenging. Nevertheless, the existing simulators focus mainly on urban driving. Therefore, in this work, we describe our approach to building a flexible real-time testing platform for unmanned ground vehicles for indoor and off-road environments. Our platform consists of our original simulator, robotic operating system (ROS), and a bridge between them. To enable compatibility and real-time communication with ROS, we generate data interchangeable with real-life readings and propose our original communication solution, UDP Bridge, that enables up to 9.5 times faster communication than the existing solution, ROS#. As a result, all of the autonomy algorithms can be run in real-time directly in ROS, which is how we obtained our experimental results. We provide detailed descriptions of the components used to build our integrated platform.

## 1. Introduction

Although autonomous vehicles are considered to shape the future of mobility, their perception of the surrounding world and vehicle control remain major challenges in robotics. This is heavily influenced by the fact that autonomous vehicles are extremely heterogeneous and must operate in diverse and complex environments with many external factors. Despite the high complexity of urban environments, it is the non-urban areas that are more challenging—the structuring characteristic of cities cannot be seen here, and the behavior of the vehicle must be deduced without specific prior assumptions.

Autonomous systems can use data from multiple sensors to understand the surrounding world, e.g., LiDARs (light detection and ranging), cameras, IMUs (inertial measurement units), odometry and GNSS (Global Navigation Satellite System). In order to find a proper system configuration, thorough testing is needed (via real-world testing, pre-recorded datasets or simulations [1]). Testing in real-life and using pre-recorded data require large investments of time and funds, and covering all necessary scenarios is difficult or even impossible, e.g., in safety-critical domains and inaccessible environments. As an alternative, simulations can be used. The available works and simulators focus mainly on visual perception algorithms and testing four-wheel personal cars in urban landscapes, while tracked vehicles and basic autonomy algorithms, such as simultaneous localization and mapping (SLAM), are largely overlooked. Therefore, it is vital to deliver flexible simulation frameworks that can be used to test different autonomy algorithms for diverse environments and vehicles.

In this work, we describe our approach to building a real-time testing platform for unmanned ground vehicles (UGV) for indoor and off-road environments. Our platform consists of our original simulator (graphical and physical capabilities of vehicles and an environment), robotic operating system (ROS), in which all of the autonomy algorithms operate (the same as on real vehicles), and a dedicated bridge between them. We propose an original solution, UDP Bridge, that enables efficient communication between the simulation and ROS. It is up to 9.5 times faster than Siemens’ solution, ROS# [2]. Raw simulated sensor readings are sent to ROS; thus, data are interchangeable with real-life readings. We present the capabilities of our platform for testing of two SLAM algorithms in indoor and non-urban environments and compare the results obtained on a simple real-life and simulated track. Using our UDP Bridge allowed us to perform all of the experiments in real-time directly in ROS. We provide the detailed descriptions of the methods, components and parameters used to build our integrated platform. We believe that the article can serve as a guideline to build such platforms and.or to improve the existing ones.

## 2. Related Work

Simulation environments range from simple environments, such as Player and Stage (2D) or Gazebo (3D) with significantly limited visual and physical capabilities [3] to more advanced solutions that emerged due to the dynamic development of autonomous cars. Among them, CARLA [4] seems to be the most powerful one (numerous sensors, urban layouts, vehicles and dynamic actors). A competing solution is Microsoft AirSim [5], which supports aerial vehicles and one car model with far less sophisticated sensor simulations (e.g., without sensor-specific noise). The third simulator focused on urban environments is the commercially licensed LG SVL simulator, which is no longer supported. Research works (e.g., [6]) focus mainly on the generation of labeled LiDAR point clouds for object recognition tasks, while other work [7] aims to do the same with camera data for planetary robotics. Out of the described solutions, the most similar framework to ours is CARLA. Nevertheless, it does not support vehicles with differential driving, although the great majority of land platforms use it (only those with Ackermann-type steering are available). Additionally, CARLA uses the OpenDRIVE format (characteristic for public roads) to describe the environment, which is not suitable indoors or off-road. In contrast to other works, which focus mainly on visual perception algorithms, our simulator focuses mainly on tasks such as localization, mapping and path determination. In our previous works, we verified the accuracy of our LiDAR data generation procedure [8].

To enable a simulation’s compatibility with ROS, communication is required. ROS itself provides a suite that can be used to build such solutions—Rosbridge [9]. It is used by numerous works: [10] extends it to support mixed-reality simulation for delivery robots and work [11] to create a monitoring solution for industrial processes. Another solution based on Rosbridge is a set of software libraries created by Siemens, ROS# [2]. It uses a WebSocket protocol (based on transmission control protocol—TCP) and ROSBridge to send data between ROS and Unity. Additionally, the developers of different simulators deliver dedicated communication solutions for their systems—e.g., CARLA ROS Bridge [12]. Ref. [13] presents a communication method for a different application than the one descibed in our paper—a cloud-based virtual reality platform. To enable communication between ROS and Unity in their solution, they created a cloud-based architecture to interface ROS with Unity. Additionally, in work [14], the authors presented an interface dedicated to virtual reality and augmented reality teleoperation, ROS Reality. Their package allows for a robot using ROS to bilaterally communicate over the Internet with the Unity game engine based on the WebSocket protocol. ROS2Unity3D [15] is a method that can be used to connect eROS and the Unity3d engine. The authors use ZeroMQ, Google Protobuf and GStreamer to build it. ZeroMQ is broker-less, open-source, and supports routing capabilities. Similarly to ROS#, the solution uses the TCP/IP protocol, but they managed to make the communication more efficient by, among other things, reducing the single packet size. In contrast to other works, we do not use ROSBridge, WebSocket, or the TCP protocol, but we propose our original solution based on the User Datagram Protocol (UDP) for efficiency (it is more suitable than TCP for real-time streaming services [16])—UDP Bridge.

## 3. Simulation Overview

The simulator uses the Unity rendering engine based on the DirectX 11 framework, supports shaders and uses the nVidia PhysX physics engine for real-time calculations. This allows the use of advanced graphic effects, making the simulation highly realistic. We used 3D modeling to create the simulation elements and dedicated C# scripts to replicate their functionalities (e.g., sensors). We ensured full transparency between data from physical devices and virtual devices. We used UDP Bridge (described in Section 4) to connect our simulation to ROS. We present the system environment used for the simulation setup in Figure 1.

### 3.1. Building an Environment

Different approaches can be used to create an environment: *computer design* (manually creating a model), 3D *laser scanning* and *photogrammetry* (based on subsequent photos of an environment). Unity provides many assets and allows modeling and loading models created, e.g., in Blender. To create indoor environments, we prepared 3D models of the room and added texturing to increase the realism. Similar mechanisms apply to modeling the outside world. Using standard methods, objects for a virtual map representing any area can be created. In the case of large virtual areas representing real terrain profiles, it is practical to use terrain *heightmaps* (see Figure 2 for an example of a 15 km by 15 km area in Poland).

### 3.2. Building a Ground Vehicle Model

A vehicle model is defined as a graphical and a physical model, which in combination enable the operation of virtual sensors, platform control, and a view in simulation. A graphical model (prepared using any of the modeling methods discussed in Section 3.1 and a texture) is used for visualization (e.g., for verification of sensors’ placement). Unity uses the PhysX physics engine to create kinematic and dynamic models as needed. In this case, it was crucial to prepare a dynamic model due to the fact that the control of the vehicle should be done directly through commands generated in ROS. We considered the vehicle rigid body motion in 3D space to have 6 degrees of freedom (x, y, z, roll, pitch, yaw). Defining a dynamic model involves specifying physical parameters (mass, friction, drag, elasticity, damping coefficients) for all rigid bodies that can be isolated within a vehicle and defining their collision models to enable collision detection and simulation of physical interactions between them. The joints of subsequent solids comprising the vehicle and their parameters (range of motion, mass, speed limits and acting forces) were also specified. The sensors described in Section 3.3 can be embedded in the models.

### 3.3. Building Sensor Models

In this section, we describe the details of individual sensor simulations. Our testing platform provides simulations for many types of sensors of any specification. Details regarding the presented sensors are discussed in the following subsections.

#### 3.3.1. LiDARs 2D and 3D

LiDARs are key devices for autonomous vehicles. Their simulation requires an efficient mechanism determining the distance to the nearest objects. In one of our previous articles, [8], we described in detail the simulation procedure of LiDAR data. The basic LiDAR parameter is the number of channels (vertical resolution). Other parameters refer to the horizontal (up to 360${}^{\circ }$) and the vertical (for 3D LiDARs) field of view, as well as the angle between successive measurements of the rotating lasers (horizontal resolution) and the field of view in a discrete period of simulation time (the angular resolution per data block in a real device). We also simulate measurement errors based on the parameters provided by the manufacturer. Additionally, the rolling shutter effect simulation should be included. In the article [8], we described the impact of this effect on positioning/localization accuracy and error propagation over time. In Figure 3, we present an example scan of the real and simulated laboratory room using Velodyne VLP-16 and its simulation (in [8], we have included a table with parameters of real and simulated Velodyne VLP-16 LiDARs). The LiDAR VLP-16 and its equivalent in our simulation is defined by its horizontal and vertical field of view and resolution (30${}^{\circ }$ and 360${}^{\circ }$, 0.1${}^{\circ }$ and 2${}^{\circ }$, respectively), the number of rotations per second (considered 10 Hz), the angular resolution per data block (2.4${}^{\circ }$ and 15${}^{\circ }$ in the simulation) and the accuracy of a single measurement (0.03 m).

#### 3.3.2. Cameras

A monocular color, stereovision and depth cameras can be defined in our simulator. To simulate a monocular color camera, we define the image size, quality, distortions (caused by matrix/lens imperfections in actual devices), resolution, frame rate, compression quality and field of view of the camera. The Unity rendering engine supports shaders and post-processing of image frames; therefore, distortions characteristic of digital cameras can be introduced: grain, chromatic aberration, barrel or cushion distortion, vignetting and focus depth. To simulate a stereovision camera, two monocular cameras have to be prepared and arranged according to the actual camera design. To simulate the depth camera, a depth buffer (with a Z coordinate for each pixel) generated by the rendering engine was used to display the 3D scene from the perspective of the virtual camera. The effect is achieved via the shader program, which is able to read information from the depth buffer (see the example in Figure 4). The same parameters as for the monocular camera are defined, and additionally, the minimum and maximum distances at which the depth are determined. The simulation model was developed based on a stereovision camera with a given resolution (HD—1280 × 720) and a field of view (vertical—72${}^{\circ }$ and horizontal—104${}^{\circ }$).

#### 3.3.3. Inertial Measurement Units

In the simulation, we used an IMU consisting of a gyroscope and an accelerometer. With the definition of the acceleration vector as the second derivative of distance over time, the change in the position of the simulated object between two points during the time period corresponding to the frequency of the actual sensor was determined. Based on the determined velocity and the previous value, the acceleration was calculated. Then, we took into account the value of the standard acceleration due to gravity. Similarly, the angular velocity of the object was determined. This was supplemented by the quaternion of the object’s rotation, which was read directly from the simulation. Then, all the values were subject to a measurement error (a pseudo-random number from the range of values provided by the manufacturer of a given sensor model). The error value was calculated as a pseudo-random number with a normal distribution determined by the Marsaglia polar method [17]. The maximum data update rate depends on the physics engine simulation step, which was set to a fixed value of 2.5 ms. This corresponds to a frequency of 400 Hz (compatible with high-end IMUs). The developed IMU model also supports the generation of data with a larger time interval (for different actual models).

#### 3.3.4. Odometry Sensors

Odometry is an incremental method of vehicle localization usually based on data from a sensor placed inside the vehicle wheel. By integrating the distance travelled, the position of the vehicle is measured and determined. Odometry data typically consists of two components: position and speed vectors, as well as, optionally, error covariance matrices. These values can be calculated based on readings from encoders counting engine revolutions (e.g., brushless DC motors with Hall effect sensors). Electrical pulses are counted and stored in 64-bit counters (input for the odometry algorithm). In our solution, we simulate the encoder counters by calculating the angle difference between the wheel rotation in successive steps of the physics engine simulation. When the difference reaches a value compatible with the angular resolution of the real encoders, the particular wheel’s counter is incremented or decremented (based on movement direction), and the counter value is sent to the autonomy system at a frequency compatible with the real-life controller—the odometry data can be calculated by the same algorithm that is used by an actual robot.

#### 3.3.5. GNSS

In GNSSs, receivers on the ground determine the distance to individual satellites by measuring the time of radio signal arrivals from them. The signal contains information on the satellites’ position, their theoretical trajectory and a transmission timestamp. They are used to determine the receiver’s position in the longitude, latitude and altitude (LLA) format, and then converted to the chosen reference system. To simulate the GNSS readings, we used mathematical transformations between different reference systems. We defined an LLA format reading and converted it (using WGS-84 standard parameters) to the ECEF (Earth-centered, Earth-fixed) format. Each update of the virtual data requires updating the ECEF position and reconverting it to the LLA format. We subjected measurements to an error by adding pseudo-random values with a normal distribution corresponding to the measurement accuracy to the coordinates before the update. We included the GNSS data update rate parameter (for real receivers, it usually oscillates between 1 Hz and 5 Hz).

## 4. Efficient Communication with UDP Bridge

To test autonomous solutions in ROS via simulation in real-time, efficient communication is necessary. One of the existing Unity-ROS communication solutions is ROS# [2], implemented by Siemens. It uses a WebSocket protocol (an application layer protocol in the ISO/OSI model based on TCP) and JSON files. Unlike binary data, the transmission of data using a JSON file causes significant redundancy and limits the maximum amount of data to be sent at once (problematic especially for virtual camera or LiDAR data transmission). Moreover, the TCP protocol results in significant time overheads that occur when establishing a connection, maintaining a session, or retransmitting. This limits the applicability of ROS# in real-time testing.

For that reason, we designed and implemented our own communication solution—UDP Bridge. We chose the User Datagram Protocol (UDP) to build it. In Figure 5, we present an overview of our platform. UDP Bridge offers bidirectional simulation-ROS communication by developing libraries for both ROS and the NET platform. Due to the possibility of sending large amounts of data from the simulator, it was decided to use UDP, as it is more suitable than TCP for real-time streaming services [16] (it minimizes the time overhead inherent in TCP). We compare the efficiency of ROS# and our UDP Bridge in Section 6.1.

The UDP header is limited to 4 bytes (destination and source port numbers, the packet length, and the checksum). This allows the maximum amount of data to be transmitted in a single datagram. Data sent over the UDP Bridge have an additional variable-length header that contains a number of elements necessary for the full integration of virtual data into ROS and effective two-way communication. Table 1 shows the structure of the additional data header, which has a minimum length of 20 bytes. Each message type has a specific index based on which data in a datagram outside the header part is parsed. UDP Bridge allows to send many types of messages supported by ROS, including *PointCloud2*, *Imu*, *CompressedImage*, *NavSatFix* and *WheelEncoders* (for sensors described in Section 3.3) and *Range* (for distance measurements).

The header also contains the name of the ROS topic on which data are to be published or the name of the appropriate connection in the transformation tree. In addition, UDP Bridge enables the exchange of data of other types and allows the implementation of user applications. The following information types can be used: Subscribe (a control packet with a request to subscribe messages on a given ROS topic), Clock (time used to synchronize consecutive measurements in the simulator), Twist (two vectors with linear and angular velocity acting as standard input for the motor controller or physical model in the simulator), PoseStamped (position in space and time—a robot’s position relative to the map), and OccupancyGrid (a two-dimensional environment map with obstacles. Together with PoseStamped, it allows to visualize a vehicle at a specific map location).

The UDP Bridge is a multithreaded program written in C++ as a ROS node. A wrapper for the .NET platform and the Unity engine is also implemented. In Algorithm 1, we provide a pseudocode of the thread dedicated to receiving and sending data using a UDP socket. Algorithm 2 presents the second thread that processes the data before publishing them in ROS and obtains data from subscribed topics.
**Algorithm 1:** The thread for receiving/sending data.
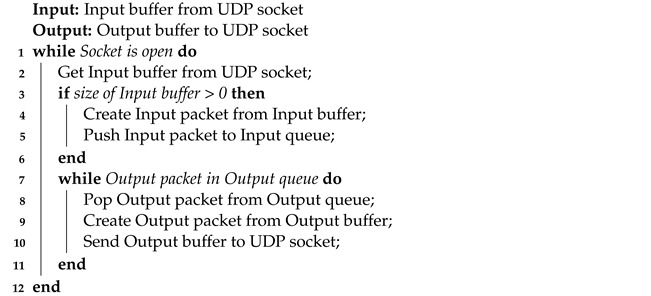


**Algorithm 2:** The thread for processing data.

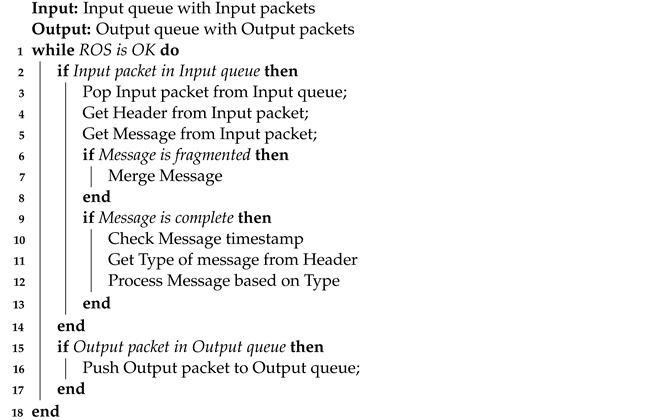



## 5. Experimental Setup

For the purpose of this study, we have performed the following experiments:Comparison of communication efficiency of ROS# and UDP Bridge;Comparison of Lidar-based SLAM error values in reality and simulation;Comparison of true and estimated trajectories in an indoor environment;Comparison of true and estimated trajectories in an outdoor environment.

To verify the communication efficiency, we sent different amounts of data between our simulator and ROS using UDP Bridge and ROS#. For comparison of errors in real-life and simulation, we used Google Cartographer SLAM [18] on a simple real track and the corresponding simulated one (see Figure 6). We also chose two SLAM algorithms, LiDAR-based SLAM (Google Cartographer [18]) with a monocular camera and feature-based SLAM (ORB SLAM2 [19]), and examined their performance (mapping and estimated trajectories) for a decontamination robot in an indoor environment (see Figure 7) and a tank in an outdoor environment (see Figure 8). For both environments, we use closed tracks to examine whether issues with loop closure occur.

Our indoor robot uses 2D LiDAR, IMU and odometry. In the case of an outdoor environment, it uses a 3D LiDAR and IMU. ORB SLAM2 uses only the monocular camera in both examined scenarios. Due to compatibility with ROS and our UDP Bridge communication, we performed all of the experiments in real-time and used an ROS visualization tool, RViz, to obtain the results (we projected them on the simulated environments). Our results show that although UDP transmission can encounter some packet loss, errors, reordering or duplication, it can be successfully used to build a real-time simulator of autonomous robots.

## 6. Experiments

The results in Section 6.1 prove that our solution is significantly faster than ROS#. The results in Section 6.2 show that the obtained SLAM errors in real life and in simulation are very similar; such a verification procedure can be used for evaluation of simulation frameworks. The results considering Google Cartographer and ORB SLAM2 in indoor and outdoor environments are presented in Section 6.3 and Section 6.4, respectively. The experiments connected to SLAM evaluation show that Google Cartographer can successfully operate in real-time on data exchanged in our simulator via the UDP Bridge.

### 6.1. Comparison of Performance of Communication via UDP Bridge and ROS#

The aim of our simulation is testing in real-time; thus, crucial aspects of our system are efficiency and fast communication. In Figure 9, we compare the performance of two simulation-ROS communication solutions, ROS# and our solution, UDP Bridge. It is clearly visible that UDP Bridge significantly outperforms the ROS# solution in all of the examined cases (from 2 times shorter transmission at minimum for smaller amounts of data up to 9.5 times for larger ones). For all the examined cases, in which we sent a minimum of 10 MB, the UDP Bridge is around 9 times faster than ROS#.

### 6.2. Comparison of LiDAR-Based SLAM Performance in Real and Simulated Environments

We used the real and simulated tracks presented in Figure 6 to examine the Google Cartographer SLAM algorithm accumulated error (determined as a sum of squared distances [mm] between the estimated and true poses in the examined checkpoints). The robot drove the track back and forth (30 m in total). In Figure 10, we have presented how the error increases with the travelled distance. It is visible that the results are very similar for real and simulated data. Such a comparison is one of the ways that can be used to verify the accuracy of simulated data while creating a simulator.

### 6.3. Comparison of Trajectories in an Indoor Environment

The results of our experiments performed in a simulated indoor environment presented in Figure 7 have been shown in Figure 11. It can be observed that both the examined algorithms have obtained a high mapping performance—the most prominent objects have been reflected by Google Cartographer, and the features of these objects have been found by ORB SLAM2. The trajectory estimated by Google Cartographer is much closer to the ground truth than the one estimated by ORB SLAM2 (issues at the corners can be observed). Nevertheless, on a straight path, both examined SLAM algorithms are accurate. The results show that to improve the performance on the corners of monocular SLAM, it would be beneficial to use additional sensors: IMU or odometry. Nevertheless, both SLAM algorithms can handle this environment to some extent.

### 6.4. Comparison of Trajectories in an Outdoor Environment

The results of our experiments performed in an outdoor environment presented in Figure 8 are shown in Figure 12. It can be observed that the mapping performance of both algorithms is relatively accurate; the buildings have been reflected well, while only some trees have been represented. At the end of the track, it can be observed that the Cartographer’s map is slightly rotated. Both estimated trajectories are very close to the ground truth. ORB SLAM2 outperformed Google Cartographer, which exhibited some issues with loop closure. Nevertheless, it is visible that a vehicle using both the examined SLAM algorithms can successfully operate in the simulated environment.

## 7. Discussion

In this section, we discuss some of the advantages and limitations of testing a robot in a simulated environment and using the UDP Bridge for simulation. We also provide the examples of some tools that can be used for testing in simulations (simulators, experiments and visualization).

Advantages and limitations of the approach. The advantages of simulation testing can be: (1) performing more tests at lower cost is possible. In the design phase, it is not necessary to purchase sensors to test all the possible configurations; (2) easy access to precise ground truth; (3) the possibility of conducting real-time experiments under identical conditions; (4) lower cost than real-life testing; and (5) security—no real damage is done in the case of a robot accident. On the other hand, we can highlight some disadvantages: (1) it is time-consuming to build a very accurate environment; (2) although modern simulators offer high-quality graphics and physics, the simulated world is always a simplified version of the real one. When it comes to UDP Bridge, the most obvious advantages are its efficiency and simplicity of the solution and its easy replication. The disadvantages can be the lossy nature of the UDP protocol. It can have more of an impact on the system when the simulation is held on one computer and ROS is held on another machine—here, the physical communication channel can also impact the losses. In the case of hosting both of these components on one machine, it is less impactful.

Tools. Different tools can be used for the simulation of AGVs, from the simpler ones such as *Stage* or *Gazebo*, to some more advanced ones: e.g., *CARLA* [4] or *Microsoft AirSim* [5]. Microsoft AirSim is dedicated to aerial platforms but offers one car model. For testing autonomous vehicles in urban environments, CARLA seems to be the most suitable simulator. It provides different urban layouts and dynamic actors. CARLA does not support differential steering; thus, to simulate such land platforms, it is better to build a dedicated simulator based on graphics engines, e.g., Unity (the same as the authors of this paper). When it comes to experiments and visualization, ROS offers its own visualization component, RViz [20]. Alternatively, one can use some more advanced tools, e.g., Webviz [21] or Foxglove [22].

## 8. Conclusions and Future Work

In this paper, we have described our approach to building a real-time testing platform for unmanned ground vehicles and presented the results regarding LiDAR-based and monocular SLAM algorithms (Google Cartographer and ORB SLAM2) tested in two diverse environments: a laboratory room (representing an indoor environment) and rural scenery (representing a non-urban outdoor environment). We have proposed an efficient communication solution that connects the simulation with ROS enabling autonomy algorithms testing directly in ROS in real-time. In our experiments, we have presented that our solution significantly outperforms a ROS# solution. In all of the experiments, data (regardless of the sensors used) could be seamlessly exchanged between the simulator and ROS, and Google Cartographer could successfully operate. We believe that our paper can be used by other researchers as a baseline to create their own ROS-compatible testing platforms and to extend the existing solutions.

## Figures and Tables

**Figure 1 sensors-22-08493-f001:**
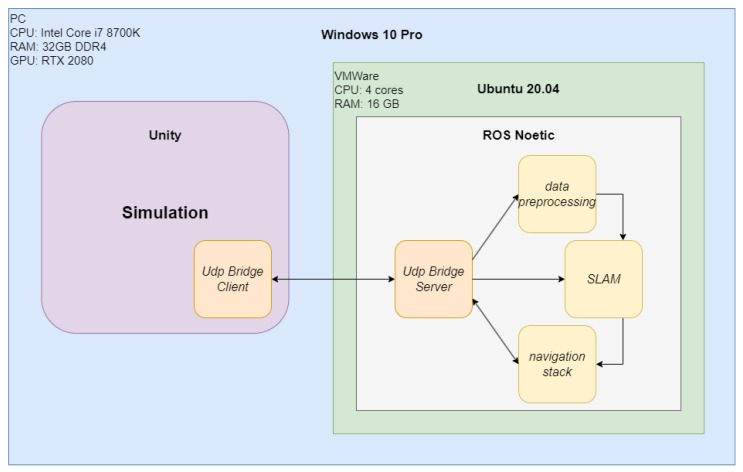
The system environment used for simulation setup.

**Figure 2 sensors-22-08493-f002:**
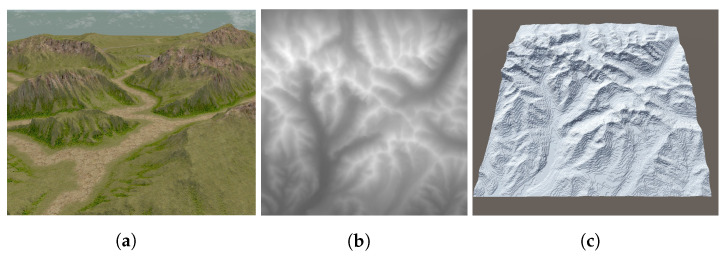
Building a simulated environment (**a**) from a terrain heightmap (**b**) and 3D model (**c**).

**Figure 3 sensors-22-08493-f003:**
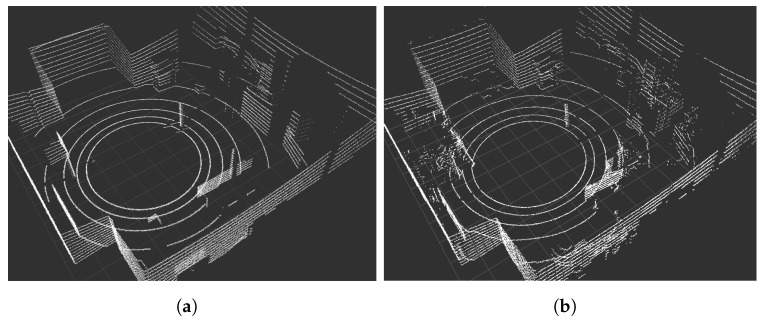
Velodyne VLP-16 point clouds obtained via simulation (**a**) and in real life (**b**).

**Figure 4 sensors-22-08493-f004:**
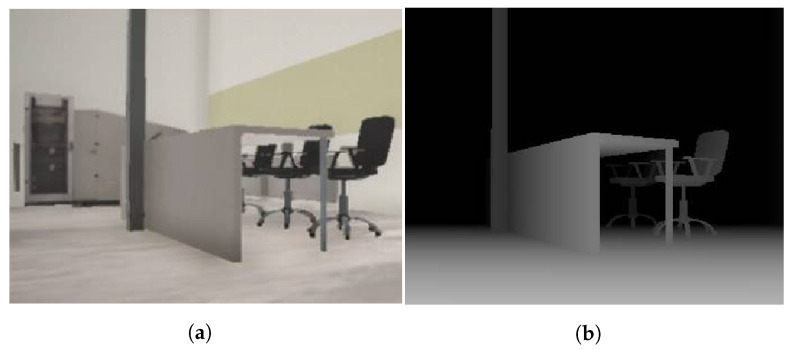
Simulated environment (**a**) and the simulated depth camera output (**b**).

**Figure 5 sensors-22-08493-f005:**
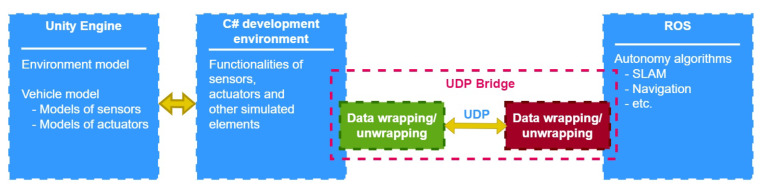
Using UDP Bridge to enable communication between Unity and ROS.

**Figure 6 sensors-22-08493-f006:**
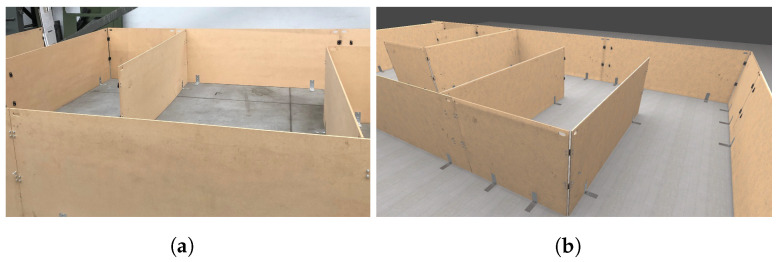
Real (**a**) and simulated (**b**) tracks used in the experiment.

**Figure 7 sensors-22-08493-f007:**
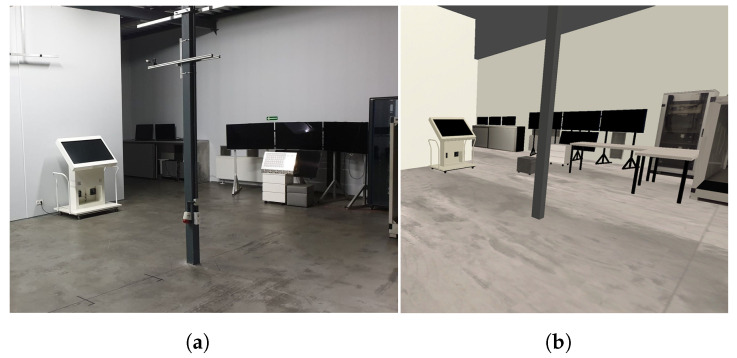
Real-life (**a**) and the corresponding simulated (**b**) room.

**Figure 8 sensors-22-08493-f008:**
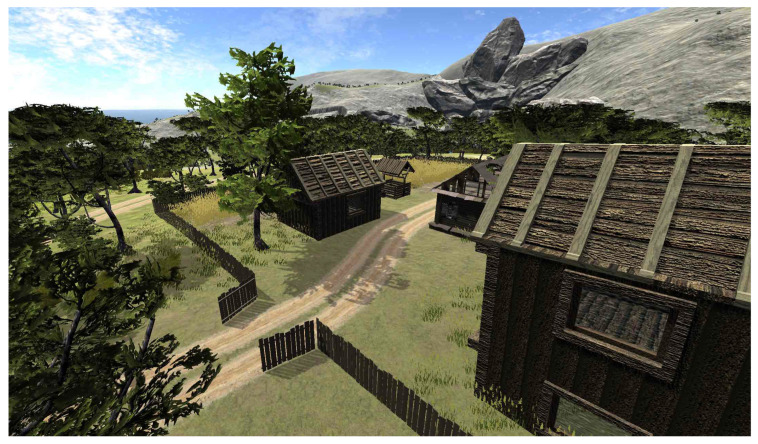
Simulated rural environment.

**Figure 9 sensors-22-08493-f009:**
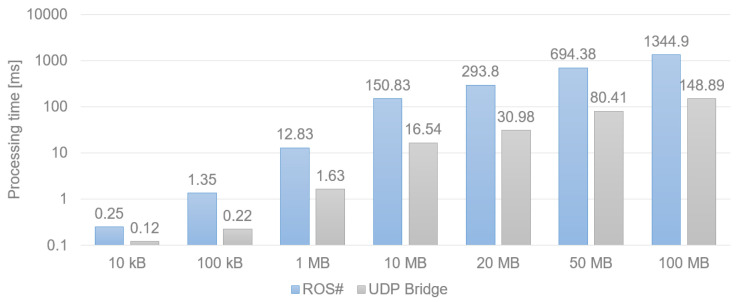
Time in milliseconds needed for data transfer via ROS# and UDP Bridge.

**Figure 10 sensors-22-08493-f010:**
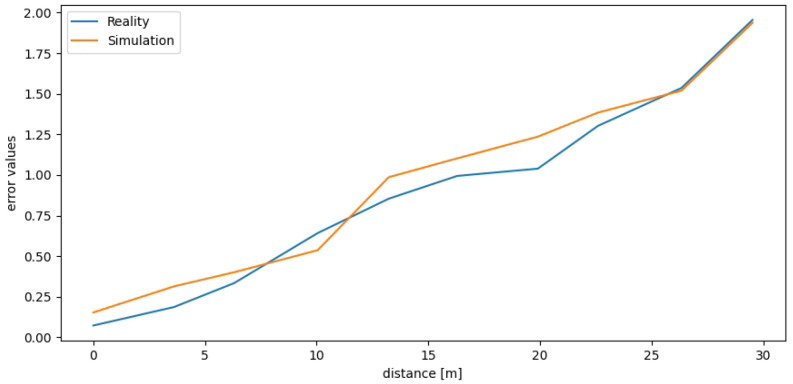
Real and simulated LiDAR-based SLAM error values.

**Figure 11 sensors-22-08493-f011:**
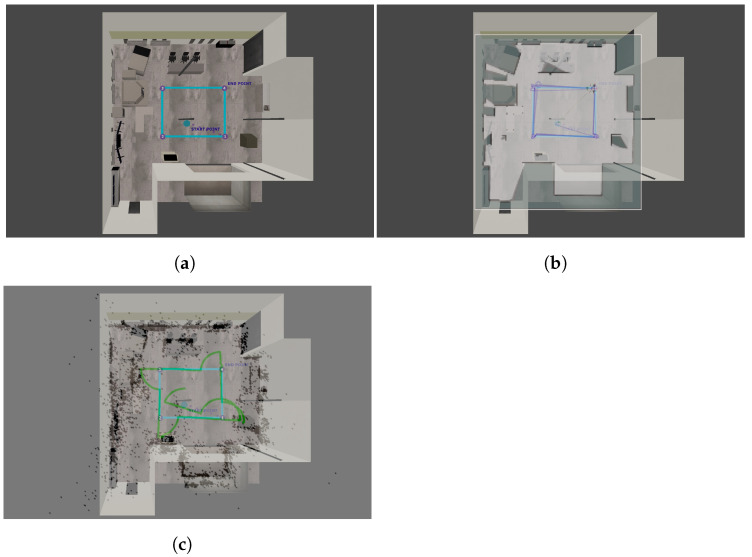
True (**a**) and estimated trajectories using Cartographer (**b**) and ORB-SLAM2 (**c**) in the indoor environment.

**Figure 12 sensors-22-08493-f012:**
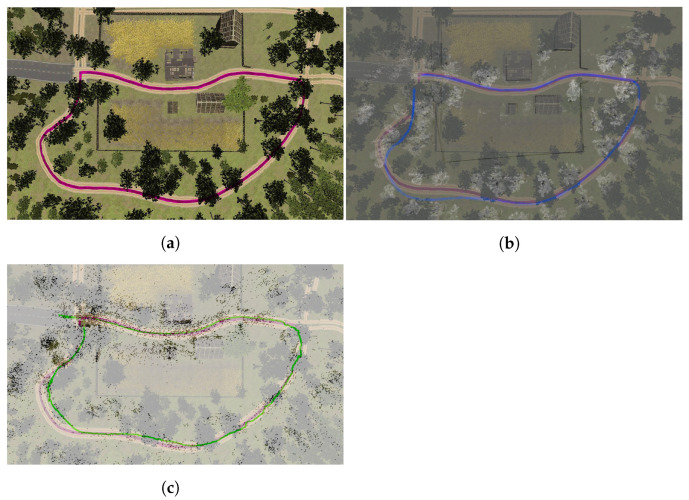
True (**a**) and estimated trajectories using Cartographer (**b**) and ORB-SLAM2 (**c**) in the outdoor environment.

**Table 1 sensors-22-08493-t001:** Structure of the UDP header used by our UDP Bridge.

File Name	Data Type	Size	Description
MsgType	int8_t	1	Type of message
HeaderSize	int8_t	1	Size of the header in bytes
SeqNumber	int32_t	4	Message sequential number
PartNumber	int8_t	1	Message sequential number
PartsMax	int8_t	1	Number of all message fragments
TimeSec	int32_t	4	Time in POSIX format (integer part)
TimeNsec	int32_t	4	Time in POSIX format (fraction part)
TopicSize	int8_t	1	Size of topic name field
LinkSize	int8_t	1	Size of link name field
Topic	string	TopicSize	Topic name
Link	string	LinkSize	Link name (in a transformation tree)
Padding	-	0/1/2/3	Alignment of the header length

## Data Availability

Not applicable.

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
