# Peer review of "Building a Real-Time Testing Platform for Unmanned Ground Vehicles with UDP Bridge"

_sensors, 2022, doi:10.3390/s22218493_

Round 1

Reviewer 1 Report

The authors proposed UDP bridge to communicate between simulation environments and ROS. The proposed method is faster compared to ROS. The study is interesting in terms of data transfer between ROS and simulation.

There are some points that need to be clarified to improve the manuscript:

1. References/related work

The authors need to cite more references/methods in related studies concerning the data-transfer method between the simulation environment and ROS. What is different among the methods?

2. Ground Vehicle model

How many degrees of freedom (DOF) do you use?

Do you implement dynamic or kinematics modeling in your ground vehicle?

3. Inertial Measurement Units (IMU)

The authors mentioned using an accelerometer and gyroscope.

Do you use a magnetometer in your model?

How do you model the noise generated by the accelerometer and gyroscope?

4. Figure 5

The y-label is hard to follow. What do the authors mean by [MS], is that millisecond?

Author Response

We would like to thank the reviewer very much for spending a significant amount of time to provide us with many constructive and helpful comments. We tried to fulfill all of the given suggestions in our revision, our response has been added as the attachment.

Reviewer 2 Report

The authors have developed a testing platform for unmanned ground vehicles. After reviewing the manuscript, a few concerns are addressed.

1. The authors need to provide the specifications of the sensors incorporated in the testing platform.

2. The authors need to provide the system environment used for simulation setup.

3. The authors could provide the existing testing platforms for reader's interest and the proposed work significance.

4. Providing the discussion section to highlight the advantages and limitations of their proposed work will give more interest to the reader.

Author Response

We would like to thank the reviewer for the time and effort sacrificed to provide us with many constructive and helpful comments. We tried to cover all of the points of the review in our revision. We hope that the reviewer’s concerns have been eliminated. Our response has been added as the attachment.

Round 2

Reviewer 2 Report

After reviewing the revised manuscript, it is observed that the authors have addressed most of the concerns highlighted earlier. 

I would like to recommend this revised manuscript towards acceptance in this journal, If the editor has no concerns.